# The Effect of *Schisandra chinensis* Baillon on Cross-Talk between Oxidative Stress, Endoplasmic Reticulum Stress, and Mitochondrial Signaling Pathway in Testes of Varicocele-Induced SD Rat

**DOI:** 10.3390/ijms20225785

**Published:** 2019-11-17

**Authors:** Keshab Kumar Karna, Bo Ram Choi, Min-Ji Kim, Hye Kyung Kim, Jong Kwan Park

**Affiliations:** 1Department of Urology, Institute for Medical Sciences, Chonbuk National University Medical School-Biomedical Research Institute and Clinical Trial Center of Medical Device, Chonbuk National University Hospital, Jeonju 54907, Korea; karnakeshab@gmail.com; 2Jaseng Spine and Joint Research Institute, Jaseng Medical Foundation, Seoul 135-896, Korea; bohosun@hanmail.net; 3College of Pharmacy, Kyungsung University, Busan 48434, Korea; xyuuuu33@ks.ac.kr

**Keywords:** *Schisandra chinensis* Baillon, varicocele, endoplasmic reticulum stress, oxidative stress, cytokines, mitochondria, testis, apoptosis

## Abstract

*Schisandra chinensis* Baillon (SC) has been utilized for its antioxidants and anti-inflammatory activities in a broad variety of medical applications. However; SC uses for improving fertility in males and related disorders with proper scientific validation remain obscure. The present study aimed to investigate the effects of SC on varicocele (VC)-induced testicular dysfunction and the potential molecular mechanism associated with VC-induced germ cell apoptosis. The male Sprague–Dawley rats were equally divided into four groups consisting of 10 rats in a normal control group (CTR), a control group administered SC 200 mg/kg (SC 200), a varicocele-induced control group (VC), and a varicocele-induced group administered SC 200 mg/kg (VC + SC 200). Rats were administrated 200 mg/kg SC once daily for 28 days after induction of varicocele rats and sham controls. At the end of the treatment period, body and reproductive organ weight, sperm parameters, histopathological damages, proinflammatory cytokines, apoptosis markers, biomarkers of oxidative stress, endoplasmic reticulum (ER) stress, and steroidogenic acute regulatory protein (StAR) were evaluated. The effects of SC extract on human sperm motility were also analyzed. SC treatment reduces VC-induced testicular dysfunction by significantly increasing testicular weight, sperm count and sperm motility, serum testosterone level, Johnsen score, spermatogenic cell density, testicular superoxide dismutase (SOD), glutathione peroxidase (GPx) and catalase level, and steroidogenic acute regulatory protein (StAR) level. Furthermore, the effects of SC on malondialdehyde (MDA) level, reactive oxygen species (ROS)/reactive nitrogen species (RNS) level, apoptotic index, serum luteinizing hormone (LH) and follicle stimulating hormone (FSH) levels, Glucose-regulated protein-78 (Grp 78), phosphorylated c-Jun-N-terminal kinase (p-JNK), phosphorylated inositol-requiring transmembrane kinase/endoribonuclease 1α (p-IRE1α), cleaved caspase 3, and Bax:Bcl2 in VC-induced rats were significantly decreased. Treatment with SC extracts also increased sperm motility in human sperm. Our findings suggest that the SC ameliorate testicular dysfunction in VC-induced rats via crosstalk between oxidative stress, ER stress, and mitochondrial-mediated testicular germ cell apoptosis signaling pathways. SC promotes spermatogenesis by upregulating abnormal sex hormones and decreasing proinflammatory cytokines (interleukin-6; TNF-α).

## 1. Introduction

Varicocele (VC) is defined as an abnormal enlargement of the pampiniform plexus of the spermatic cord. It is the leading cause of infertility in men [1]. The incidence of VC in the healthy male population is around 15% to 20%. VC occurs in 35–50% of men with primary infertility and 80% of men with secondary infertility [2]. Varicocelectomy is the most commonly used technique to treat VC-induced male infertility. However, after varicocelectomy treatment for VC-induced infertile men, studies have reported 50–80% improvement in semen parameters with a pregnancy rate of only 31–71%, indicating that varicocelectomy cannot completely restore fertility [3]. VC is ordinarily accompanied by decreases in testicular volume, sperm count, sperm motility, Leydig cell function, and testosterone level [4]. Although the pathophysiology of varicocele-related infertility is not completely understood yet, it is associated with multifactorial mechanisms. The mechanism of VC-induced testicular dysfunction includes increased temperature, inflammatory cytokines, oxidative stress, endoplasmic reticulum (ER) stress, hormonal imbalance, and apoptosis [5,6,7]. 

A man with idiopathic infertility is associated with higher seminal reactive oxygen species (ROS) level and lower antioxidant activity [8]. Increased ROS in the testicular tissue of patients with varicocele is due to high levels of phospholipids, saturated fatty acid, and unsaturated fatty acid present in the plasma membrane of spermatozoa that make them susceptible to the release of ROS. Damaged germinal epithelium and apoptotic spermatogenic cells also increase ROS production [9]. Production of ROS has a bidirectional role in oxidative stress and ER stress [7]. Increased ROS can lead to oxidative stress that alters homeostasis in ER protein folding. Moreover, oxidative protein folding in ER can generate ROS as a byproduct, further increasing oxidative stress [10]. Cellular antioxidant mechanisms play an important role in maintaining homeostasis and reducing ROS production [10]. Various antioxidants have been reported to have beneficial effects for idiopathic male infertility. In addition, they can enhance the success rates of surgical treatment [11]. Although several empirical therapies with herbal agents for treating VC-induced infertility have been reported, none of them have been proven to be superior to others [2,9,12].

*Schisandra chinensis* Baillon (SC) is an oriental traditional Chinese medicine. SC is known to have five flavors (sour, bitter, sweet, salty, and pungent) in classical Chinese medicine and has been used for over centuries by humans [13]. In early the 16th century, SC was used in the treatment of respiratory disease, cardiovascular disease, insomnia, fatigue, and gastrointestinal disease [14]. However, pharmacological effects of SC were recognized as an official medicinal remedy in early 1960s by Russian scientists [15]. Major lignans of *S. chinensis* fruits are gomisins (A, B, C, D, E, F, G, K3, N, and J), schisandrol B, schisandrin, and schisandrin C [16]. SC possesses antioxidant, anti-inflammatory, anticancer, and antihepatotoxic activities [17,18]. SC has the ability to fortify mitochondrial antioxidant status [19]. More recently, studies have reported therapeutic effects of SC on blood sugar balance, wound healing, alleviating cough, liver injury, kidney injury, lung injury, platelet aggregation hepatitis, and cardiovascular disease [16,20]. SC has also been reported to prevent age-related disease such as osteoporosis and sarcopenia [21]. In a study on cyclophosphamide-induced dyszoospermia in male rats, *Schizandra chinensis* polysaccharide could alleviate reproductive hormones and improve fertility [22]. However, the role of SC in VC-induced testicular dysfunction and its association with oxidative stress, ER stress, and mitochondrial mediated germ cell apoptosis in VC-related male infertility remain unclear.

Thus, the objective of the present study was to explore possible protective effects of SC on male reproductive dysfunction and testicular germ cell apoptosis in VC-induced rats. In addition, this study attempted to clarify the possible molecular mechanism of VC-induced male infertility and the role of inflammatory cytokines, oxidative stress, ER stress, and the mitochondrial signaling pathway in VC-induced male spermatogenic impairment. Results of this study will lay a foundation for therapeutic potential of using SC to treat varicocele and prevent VC-induced infertility. 

## 2. Results

### 2.1. Effects of SC on Body and Reproductive Organ Weights of VC-Induced Rats

In order to investigate protective effects of SC extract on basic testicular dysfunction induced by varicocele, we established a VC-model and administrated rats with an SC extract at a dose of 200 mg/kg. There was no significant difference in body weight (sacrifice) or reproductive organ weight among all groups, except for testis weight (Table 1). Testis weight in the VC group was significantly (*p* < 0.05) decreased compared to that of the CTR group or the SC 200 group. SC 200 mg/kg treatment significantly (*p* < 0.05) increased testis weights of VC-induced rats. 

### 2.2. Effects of SC on Sperm Parameters of VC-Induced Rats

Sperm count and motility in the four groups are depicted in Table 2. Sperm count (*p* < 0.01) and motility (*p* < 0.05) in both the vas deference and the epididymis of the VC group were significantly lower than those of CTR and SC 200 groups. This result shows the deleterious effect of varicocele on sperm parameters. Pretreatment with SC 200 mg/kg significantly increased sperm count (*p* < 0.01) and sperm motility (*p* < 0.001) of VC-induced rats.

### 2.3. Effect of SC on Sperm Motility in Human Semen Sample

Total motility in patients treated with SC 0.05 mg/mL for 3 h was higher than that in the control group (Table 3). Sperm motilities in patients of the control group after 0 h of incubation with Ham’s F-1 medium were decreased compared to those after 3 h of incubation with Ham’s F-10 medium (57.8% to 50%, 67% to 65.7%, and 51.6% to 38.0% in patients 1, 2, and 3, respectively). However, they were increased in the SC 0.05 mg/mL treated group (after 0 h of incubation with Ham’s F-1 medium to after 3 h of incubation with Ham’s F-10 medium: 50% to 66.5%, 64.2% to 74.5%, and 45.8% to 48.5% in patients 1, 2 and 3, respectively). Schisandrol A (schisandrin) is major compound in SC extract (Appendix A). Sperm motility showed an increase in major compounds of SC, Schisandrol A (schisandrin) (Appendix A). SC lignans may protect spermatozoa by keeping enzymatic and antioxidant processes, such as ROS, in optimum condition.

### 2.4. SC Alleviates VC-Induced Testicular Histopathological Damage and Germ Cell Apoptosis

It has been reported that varicocele is associated with degeneration of germ cells in seminiferous tubules [7]. We performed hematoxylin and eosin (H&E) staining and TUNEL staining to observe the morphology of seminiferous tubules and apoptosis of germ cells (Figure 1). Histology analysis revealed normal morphology of seminiferous tubules in the control and SC 200 groups. All stages of spermatogonia differentiated into spermatozoa in most tubules. However, VC-induced rats showed vacuolization, irregular shape, and depleted seminiferous tubules with only spermatogonia, Sertoli cell, and few primary spermatocytes. Treatment with SC 200 mg/kg restored all morphological abnormalities in VC-induced rats. It also improved germ cell apoptosis with an orderly arrangement of multilayered epithelial cells and stages of spermatogonia differentiation into spermatozoa in tubules (Figure 1A). Furthermore, Johnsen score (*p* < 0.05) and spermatogenic cell density (*p* < 0.001) were downregulated in the VC group compared to those in the CTR group. However, VC + SC 200 group showed significant increase in Johnsen’s score (*p* < 0.05) and spermatogenetic cell density (*p* < 0.001) compared to the VC group (Figure 1B,C). To further study VC-induced apoptosis, TUNEL positive cells were counted to determine apoptotic index (Figure 1D,E). A higher (*p* < 0.001) number of TUNEL positive cells were found in VC group than that in the CTR group. SC 200 mg/kg significantly (*p* < 0.001) alleviated the increase of apoptotic index. These results suggest that treatment of SC extract improved the morphology of depleted seminiferous tubules and apoptosis in germ cells.

### 2.5. SC Reduces VC-Induced Oxidative Stress in Testes

To observe oxidative stress and protective effects of SC, we analyzed malondialdehyde (MDA), ROS/reactive nitrogen species (RNS) level, and antioxidant enzymes activity in testis tissue (Figure 2). MDA and ROS/RNS are the most critical markers of oxidative stress. Compared to CTR and SC 200 groups, MDA (*p* < 0.05) and ROS/RNS (*p* < 0.001) levels were significantly increased in the VC group. However, treatment with SC 200mg/kg significantly (*p* < 0.01) decreased their levels in VC-induced rats (Figure 2A,B). Furthermore, activities of antioxidant enzymes superoxide dismutase (SOD) (*p* < 0.001), glutathione peroxidase (GPx) (*p* < 0.05), and catalase (*p* < 0.05) were decreased in the VC group compared to those in CTR and SC 200 groups. In contrast, treatment with SC 200 mg/kg significantly (*p* < 0.05) upregulated activities of antioxidant enzymes SOD, GPx, and catalase in VC-induced rats compared to those in the VC group (Figure 3C–E). These results suggest that SC extract could improve fertility by downregulating oxidative stress in testicular tissue.

### 2.6. Effect of SC on VC-Induced Hormone Level and Anti-Inflammatory Activity

To determine whether SC extract can reverse the changes in sex hormonal levels, we used enzymatic assay to analyze serum hormones- testosterone (T), luteinizing hormone (LH), and follicle stimulating hormone (FSH)) levels in serum (Figure 3A–C). Serum testosterone level was significantly (*p* < 0.01) decreased in the VC group compared to that in the SC group (Figure 3A). However, it was not significantly lower than that in the CTR group. Serum LH and FSH levels were significantly (*p* < 0.001) increased in the VC group compared to those in CTR and SC 200 groups (Figure 3B,C). In contrast, serum testosterone level in VC + SC 200 group was significantly (*p* < 0.001) increased compared to that in the VC group (Figure 3A). Serum LH and FSH levels in the VC + SC 200 group were significantly (*p* < 0.001) decreased compared to those in the VC group (Figure 3B,C). Furthermore, we analyzed IL-6 and TNF- α to observe inflammation of testicular tissue. VC-induced rats showed marked increase in testicular levels of IL-6 (*p* < 0.01) and TNF-α (*p* < 0.001) compared to rats in CTR and SC 200 groups (Figure 3D,E). Such changes were significantly (*p* < 0.01) alleviated by pretreatment with SC 200 mg/kg. 

### 2.7. ER Stress Inhibition by SC Decreases VC-Induced Germ Cell Apoptosis in Rat Testes

We performed Western blot analysis to check ER stress protein markers for inositol requiring enzyme 1 (IRE1)-C-jun-N-terminal kinase (JNK) pathway in testicular tissues. Excessive ER stress can induce cellular apoptosis through three signaling pathways (ATF6, IRE1, and PERK) [23]. The IRE1-JNK pathway was analyzed to determine VC-induced testicular germ cell apoptosis (Figure 4A,C,D). As shown in Figure 4A,C,D, Western blot analysis revealed that ER stress chaperone molecules Grp 78, p-IRE1, and phosphorylated c-Jun-N-terminal kinase (p-JNK) were upregulated (*p* < 0.05) in the VC group compared to those in CTR group. Pretreatment with SC 200 mg/kg significantly (*p* < 0.05) decreased levels of Grp 78, p-IRE1, and p-JNK in VC rats. Furthermore, cell-specific expression of Grp 78 in testis was also analyzed by immunohistochemical staining (Figure 4A). Dark brown positive signal was detected in the VC group. In contrast, no signal of Grp 78 was detected in VC + SC 200 group (Figure 4A), indicating that SC treatment could restore testicular dysfunction in VC rats via downregulation of ER stress in rat testis.

### 2.8. Effect of SC on VC-Induced Mitochondria-Dependent Apoptotic Pathway

To determine whether the mitochondrion-dependent (or intrinsic) apoptotic pathway was involved in the action of VC-induced testicular germ cells apoptosis, crucial factors mediating the intrinsic pathway, such as Bax, Bcl2, pro-caspase-3, and cleaved caspase-3 were examined (Figure 4E–G). The VC group showed downregulation of pro-caspase-3 and upregulation of cleaved caspase-3 and Bax:Bcl2 ratio compared to the CTR group (*p* < 0.05). Treatment with SC 200 mg/kg significantly (*p* < 0.05) decreased cleaved caspase-3 and Bax/Bcl2 ration compared to the VC group. Although pro-caspase-3 level was increased in the VC + SC 200 group compared to that in the VC group, the difference between the two was not statistically significant. 

### 2.9. Effect of SC on StAR Protein of VC-Induced Rats

To assess testosterone biosynthesis, expression level of StAR protein was evaluated by Western blot and immunohistochemistry staining (Figure 4B). Expression level of StAR protein was decreased (*p* < 0.01) in the VC group compared to that in the CTR group. Pretreatment with SC 200 mg/kg significantly (*p* < 0.01) upregulated levels of StAR protein in VC rats. For confirmation, cell-specific expression of StAR in testis was determined by immunohistochemistry staining (Figure 4B). Faint expression of StAR protein was observed in Leydig cells of the VC group. In contrast, dark red signalling was detected in Leydig cells of VC + SC 200 group, suggesting that SC 200 mg/kg treatment could upregulate testosterone biosynthesis in VC rats.

## 3. Discussion

It was speculated that *Schisandra chinensis* could ameliorate testicular dysfunction from VC-induced germ cell apoptosis via inhibition of oxidative stress, the ER stress signaling pathway, and downregulation of cleaved-caspase-3 levels and Bax:Bcl2 ratio. Although several studies have previously reported anti-inflammatory and antioxidant activities of SC [24,25], the molecular mechanism and regulation of SC in antioxidant and ER stress in male infertility remain unknown. We attempted to determine the VC-induced molecular mechanism of testicular dysfunction and therapeutic effect of SC by ameliorating germ cell apoptosis.

Changes in body weight and reproductive organ weight are important parameters to analyze toxicity and effects of drugs in experimental laboratory animals [26]. In our study, VC-induced rats showed decreased testicular weights compared to rats in the CTR group. However, no difference in body weight was found among experimental groups. Moreover, hematology and clinical chemistry parameters showed no effects among all groups (Appendix A). Several studies have demonstrated decreased testicular weights in male rats with VC-induced infertility [27,28]. Pretreatment with SC significantly increased testicular weight. VC has been associated with decreased sperm count and motility [7]. In the VC group rats, sperm count and motility were downregulated compared to those in the CTR group. Decrease in sperm parameters might be due to an increase in ROS, which can damage sperm DNA and lead to apoptosis in sperm [29]. Pretreatment with SC upregulated sperm count and motility in VC rats, suggesting that SC could enhance spermatogenesis. Treatment of SC extract in human samples was consistent with animal data; further evidence of the ameliorative effect of SC extract on sperm parameters.

VC-induced rats showed a decrease of testosterone level and upregulation of FSH and LH levels. This might be due to hypothalamus–hypophysis–gonad axis [30]. Results of the present study are consistent with a previous study [30]. Treatment of SC is known to induce increased testosterone levels and decreased LH and FSH levels in cyclophosphamide-induced dyszoospermia rats [22]. Our study showed similar findings after pretreatment of VC rats with SC. In addition, our results demonstrated that VC-induced rats showed histological alteration in testes, such as impaired spermatogenesis and decreases in germ cells and vacuolization. Similar alterations have been observed in previous studies [7,31], showing that VC-induced testicular histological disruption is associated with oxidative stress and ER stress. Pretreatment of VC rats with SC preserved all histological changes, suggesting that SC has ameliorative effects on specific components of seminiferous epithelium and germ cells. Furthermore, disturbances in spermatogenic cell density and Johnsen’s score were also recorded in VC-induced rats. However, pretreatment with SC restored these disturbances. Testosterone plays an essential role in normal structural morphology and physiology of seminiferous tubules [32]. To substantiate the mechanism of germ cell apoptosis, TUNEL assays were performed. VC rats showed a significant increase in apoptotic index in seminiferous tubules. However, apoptotic index was significantly reduced after pretreatment with SC.

Oxidative stress is a well-known mechanism for testicular dysfunction in VC-induced infertility, which acts by damaging sperm membrane, proteins, and DNA known to be associated with fertility [33,34]. Results of our study showed upregulation of MDA and ROS/RNS levels in VC rats, in consensus with previous findings [7]. Antioxidant enzymes interact with free radicals and scavenge lipid peroxidation. Decrease in antioxidant activity results in excessive generation of free radicals. VC-induced rats showed significant decreases in activities of antioxidant enzymes such as SOD, GPx, and catalase in testicular tissues in the present study. However, pretreatment with SC upregulated these antioxidant enzymes’ activities and downregulated MDA and ROS/RNS in VC rats. Results from this study indicate that VC-induced oxidative stress in rat testis can be suppressed by treatment with SC. Antioxidant effects of SC extract have been previously reported [35]. Furthermore, pro-inflammatory cytokines were decreased in VC rats after treatment with SC. VC is associated with inflammatory cytokines and oxidative stress [36,37]. Under physiological conditions, inflammatory mediators play crucial regulatory roles in spermatogenesis and sperm maturation [38]. These data suggest that SC possesses anti-inflammatory functions [39].

ROS has a role in impairment of Leydig cells [26]. Decreases in testosterone levels are due to damage to the Leydig cell and StAR protein synthesis [40]. The StAR protein plays a vital role in the first and rate-limiting step to transfer hydrophobic cholesterol into the inner mitochondrial membrane, where cytochrome p450 side chain cleavage enzymes can convert pregnenolone, which is further converted by a series of enzymes to testosterone and other steroid hormones [41]. Results from our study showed significantly lower expression of StAR protein in Leydig cell of VC rats. However, SC increased StAR protein expression in Leydig cells of varicocelized rats, suggesting that SC could increase capacity of testosterone synthesis by Leydig cells. 

ER stress activation could be key signaling mechanism for germ cell apoptosis [23,42]. A previous study has reported that the ER chaperone Grp 78 is predominately present in pachytene spermatocytes, suggesting that the ER stress signaling pathway has a vital role in the process of spermatogenesis [43]. Production of ROS is interlinked with oxidative stress and ER stress. Cellular antioxidant mechanisms are crucial to scavenge and minimize ROS accumulation [26]. ER stress is initiated by activation of three signaling pathways: transcription factor 6 (ATF6), inositol requiring enzyme 1 (IRE1), and protein kinase RNA-activated-like ER kinase (PERK) [44]. PERK and IRE1 pathways predominantly lead to apoptosis in prolonged ER stress [7]. In the present study, ER stress chaperone Grp 78 was significantly upregulated in VC-induced rats. IRE1-JNK signaling pathways were presented in this study. Levels of p-IRE1 and p-JNK were significantly higher in VC-induced rats, consistent with previous findings [7]. Pretreatment with SC in VC rats downregulated the expression of ER stress sensor molecules, suggesting that cellular homeostasis mechanisms were restored.

Furthermore, the intrinsic (or mitochondrial) signal transduction pathway was studied for its role in germ cell apoptosis. Previous studies have reported that the mitochondrial signaling pathway is predominantly responsible for germ cell apoptosis in experimental varicoceles [45]. JNK plays a vital role in the translocation of Bax from cytosol to the mitochondria, either by direct phosphorylation of Bax or phosphorylation of 14-3-3, a cytoplasmic anchor of Bax; the release of cytochrome c from inner mitochondrial membrane into the cytosol; and subsequent apoptosis [26]. In the present study, levels of cleaved caspase-3 and Bax:Bcl2 ratio were significantly increased in VC groups, suggesting germ cell apoptosis via intrinsic pathways [46,47]. Taken together (Figure 5), our present results suggest that crosstalk between ER stress and the mitochondrial pathway can mediate germ cell apoptosis in VC rats. Pretreatment with SC downregulated cleaved caspase 3 and Bax:Bcl2 ratio, showing a protective role by attenuating mitochondrial apoptosis. 

## 4. Materials and Methods

### 4.1. Plant Material and Extract Preparation

*Schisandra chinensis* Baillon fruits were obtained from the Herbarium Unit of Kyungsung University (voucher number at Herbarium Unit: KHK-SC-1). Crude extract of *S. chinensis* was prepared as described previously [17].

### 4.2. Identification of Major Compounds

Major compounds of *Schisandra chinensis* Baillon (SC) were identified by HPLC-UV, respectively (Figure 6). The retention time of major peaks were compared with those in SC. The HPLC system used in the analysis was Hitachi Lachrom Elite^®^ HPLC system (Hitachi Instruments Inc., Danbury, CT, USA) equipped with an autosampler and UV detector. Chromatographic separation was accomplished on a Zorbax Eclipse XDB-C18 (250 mm × 4.6 mm, 5 μm) analytical column (Agilent Technologies, CA, USA). Water (A) and acetonitrile (B) were used as mobile phases with the gradient elution mode. The gradient elution was as follows: 0–10 min, 50% B; 10–40 min, 50–80% B; 40–45 min, 80% B. The flow rate was set at 1.0 mL/min. The injection volume was 10 μL and the UV wavelength was 254 nm. 

### 4.3. Animals and Experimental Design

Experiments using animals were performed in accordance with the care and ethics committee of Chonbuk National University Laboratory Animal Center (cuh-IACUC-2017-13, 07,08,2017), a facility with accreditation from Association for Assessment and Accreditation of Laboratory Animal Care (AAALAC). A total of 40 adult male Sprague–Dawley rats weighing between 210 and 240 g, aged 8 weeks, were supplied by KOATECH, Jeonwi-ro, Jinwei-myeon, Pyeongtaek-si, Gyeonggi-do, Korea. Rats were fed a standard rat chow diet. They were provided free access to water ad libitum. All rats were housed in groups of four per cage in the animal facility under standard living conditions (temperature: 20 ± 2 °C; relative humidity: 50 ± 10%; and light/dark cycle: 12 h/12 h). Animals were acclimated to our laboratory environment for one week. After one week of acclimatization, animals were randomly divided into four groups (ten rats/group): (1) control (CTR) group; (2) SC 200 mg/kg per oral (p.o.) group (SC 200); (3) varicocele (VC) group; and (4) VC + SC 200 mg/kg p.o. group (VC + SC 200). Left varicocele was induced for rats in VC group and VC + SC 200 group according to standard protocols [7]. Rats in CTR and SC 200 groups were sham-operated with an abdominal midline incision only. The incision was then sutured. Medication of SC was started at 30 days after varicocele induction. SC 200 was dissolved with sterile normal saline and given in a daily dose (200 mg/kg body weight; by gavage) to rats in SC 200 and VC + SC 200 groups for 28 days. Rats in CTR and VC groups received only normal saline (vehicle) during the experimental period. Rats were anesthetized with a mixture of ketamine (100 mg/mL) and 2% xylazine hydrochloride (20 mg/mL) at 48 h after the last treatment of medication. Blood samples were collected from venae cava of rats. Serum was prepared and stored at −80 °C for further biochemical criterion. Body weight and reproductive organ weight were measured. Testis tissue, epididymis, seminal vesicle, prostate, and penis were collected. They were placed in Bouin’s solution and liquid nitrogen for further analysis.

### 4.4. Chemicals and Reagents

All other chemical reagents were of analytical grade and obtained from standard commercial suppliers or as indicated in specified methods.

### 4.5. Assessment of Sperm Count and Sperm Motility

Epididymis distal cauda and the vas deferens were excised and freed from the fat pad, blood vessels, and connective tissue. The tissues were then placed in separate 1.5 mL microcentrifuge tubes, chopped into small pieces, and suspended in prewarmed normal saline at 37 °C for 5 min to dissociate and release spermatozoa. The number of motile spermatozoa and the percentage of motile spermatozoa were assessed according to method described previously [26].

Semen samples were obtained from male patients between 25 to 35 years attending our hospital for urological evaluation from October 2017 to August 2018. Approval of human study was obtained from the institutional review board (IRB) of Chonbuk National University hospital (approval number: CUH 2012-03-001-020). Those with sperm count ≥ 20 × 10^6^ and sperm motility ≥40% were included for this study. All patients provided written informed consent before study enrollment. Semen samples were collected as previously described [48]. Sperm count and motility were determined at zero hour and after 3 h of incubation with SC according to WHO recommendations. The number of motile spermatozoa, percentage of motile spermatozoa, and the increase in sperm motility were assessed according to the method described previously [49].

### 4.6. Measurements of Hormones Levels

The concentration levels of luteinizing hormone (LH), follicle-stimulating hormone (FSH), and serum testosterone were analyzed using commercially available enzyme-linked immunosorbent assay (ELISA) kits (55-TESMS-E01, mouse/rat testosterone kits, ALOCO, 26-G Keewaydin Drive, Salem, NH, USA; E-EL-R0026, rat LH Elisa kit; E-EL-R0391, and rat FSH Elisa kits; Elabscience, Houston, TX, USA) according to manufacturers’ instructions. 

### 4.7. Histopathology and Terminal Deoxynucleotidyl Transferase-Mediated (dUTP) Nick-End Labeling (TUNEL) Staining of Testis

Hematoxylin and eosin (H&E) staining was performed as described previously [7,26]. Briefly, left testis tissue was evaluated using standard light microscopy. Thirty seminiferous tubules (ST) were randomly examined in each tissue section. Spermatogenic cell density was analyzed by measuring the thickness of germinal cell layer and the diameter of seminiferous tubules. Seminiferous tubules of H&E-stained sections at magnification X400 were graded by Johnsen’s score, as described previously [26]. 

To detect apoptotic cells of left testicular seminiferous tubules, a TUNEL assays kit (Dead End^TM^ Colorimetric TUNEL System for qualitative study; Promega, Madison, WI, USA) was used according to the manufacturer’s instruction. Two repeated sections from each animal were used for quantitative analysis. In the cross section, at least 30 seminiferous tubules from each slide were counted to determine the number of apoptotic cells under a light microscope (×40 objective, Nikon, Tokyo, Japan). Apoptotic index (AI) was calculated as the percent of positive nuclei stained dark brown under a light microscope, as described previously [50].

### 4.8. Immunohistochemical Staining of GRP-78 and StAR Expression

Testis tissue sections were deparaffinized and subjected to treatment with 1x Target Retrieval Solution, pH 6.0 (DAKO, Glostrup, Denmark). Sections were then incubated with peroxidase-blocking solution (DAKO) for 15 min at room temperature (RT). After washing with 1× PBS buffer twice (5 min each), tissue sections were then incubated with rabbit monoclonal anti-Grp78 (AB 21685) and StAR (D10H12) primary antibodies at dilutions of 1:100 for 24 h at 4 °C after being blocked with serum block solution for 10 min at room temperature (DAKO). After washing again with 1x PBS twice (5 min each), slides were incubated with secondary antibody (anti-rabbit IgG; Vector Labs, Burlingame, CA, USA; catalog number: MP-7451) for 1 h, followed by reaction with AEC substrate chromogens (ImmPACT AEC Peroxidase substrate; Vector Labs, Burlingame, CA, USA; catalog number: SK-4205). After washing with deionized water for 3 min, slides were counter stained with adequate amount of hematoxylin stain (the entire tissue surface of slide was covered). Slides were then rinsed with tap water for clear background and mounted using an aqueous medium (Abcam Cambridge, MA, USA).

### 4.9. Malondialdehyde (MDA) and Reactive Oxygen Species (ROS)/Reactive Nitrogen Species (RNS) Level

Malondialdehyde (MDA) levels in testis tissue homogenates were determined using a commercially available MDA assay kit (NWLSSTM malondialdehyde assay kit; Northwest Life Science Specialties LLC., Vancouver, WA, USA) according to the manufacturer’s instructions. Absorbance of colored complex was measured at a wavelength of 532 nm by kinetic spectrophotometric analysis using a Spectra Max 180 (Molecular Devices, Sunnyvale, CA, USA). MDA concentration in the sample was analyzed by comparing the measured absorbance value to the standard curve of MDA. The level of MDA was expressed as μmole per mg tissue. ROS/RNS levels in testis tissue homogenates were determined using a fluorescence kit (STA-347, OxiSelect^TM^ in vitro ROS/RNS assay kit, Cell Biolabs, Inc., San Diego, CA, USA). Absorbance values were measured at excitation and emission wavelengths of 480 and 530 nm, respectively, with a SpectraMax Gemini XS Fluorimeter, as described previously [26]. 

### 4.10. Evaluation of Antioxidant Enzyme’s Activity

Left testis tissues (100 mg) were rinsed with 1X PBS (pH 7.4) to remove excess blood thoroughly. Activities of superoxide dismutase (SOD), glutathione peroxidase (GPx), and catalase in whole tissue supernatants were evaluated using commercial kits (item No. 706002, superoxide dismutase kit; item No. 703102, glutathione peroxidase kit; item no. 707002, catalase assay kit, Cayman Chemical, Ann Arbor, MI, USA) per the manufacturer’s instructions. Antioxidant enzymes’ activities are expressed as per milligram of protein.

### 4.11. Assessment of Inflammatory Biomarkers

Left testis tissues (100 mg) were rinsed with 1X PBS (pH 7.4) to remove excess blood thoroughly. Tissues were homogenized in 1 mL of 1X PBS with a homogenizer on ice and stored at −20 °C overnight. After two freeze–thaw cycles, the homogenate was centrifuged at 10,000× *g* for 15 min at 4 °C. The supernatant was collected and used for assays. Levels of interleukin-6 (IL-6) and tumor necrosis factor-α (TNF-α) were determined with ELISA kits (BMS625 IL-6 rat Elisa kit and BMS 622 rat TNF-α kit, Thermo Fisher Scientific, Waltham, MA, USA), following the manufacturer’s instructions. Values of IL-6 and TNF-α are expressed as per milligram protein.

### 4.12. Western Blot Analysis

Protein isolation from testis tissue and Western blot were conducted as described previously [26]. Briefly, levels of ER stress markers (glucose-regulated protein-78 (GRP-78), phosphorylated inositol-requiring transmembrane kinase/endoribonuclease 1α (p-IRE1α), and phosphorylated c-Jun-N-terminal kinase (p-JNK)), apoptosis markers (pro-caspase-3, cleaved caspase 3, BCL 2 associated X protein (Bax), and B-cell lymphoma 2 (Bcl-2)), and steroidogenic acute regulatory protein (StAR) in testis tissues were measured. A total of 30–60 μg of protein were loaded onto 8% to 12% SDS–polyacrylamide gel electrophoresis, followed by electro blotted onto PVDF membranes with a trans-blot^®^ SD semi-dry electrophoretic transfer cell (Bio-Rad, Hercules, CA, USA). Protein transfer membrane was blocked with 5% bovine serum albumin (BSA) or 5% nonfat milk for an hour at room temperature and incubated overnight at 4 °C with primary antibodies for phosphorylated antibodies (p-IRE1α (Abcam Cambridge, MA USA) and p-JNK (Santa Cruz Biotechnology, Dallas, TX, USA)) or non-phosphorylated antibodies (GRP-78 (Abcam Cambridge, MA USA), pro-caspase-3, cleaved caspase 3, Bax, Bcl-2, and StAR (Cell Signaling Technology, Beverly, MA, USA)) in 5% nonfat milk. The membrane was washed with Tris-buffered saline containing 0.05% Tween 20 (TBST, pH 7.2) three times each for 10 min prior to incubation with 1:5000 diluted secondary antibodies [antimouse, antirabbit (Cell Signaling Technology, Beverly, MA, USA)] at room temperature for 1 h. The membrane was washed three times with TBST (10 min each). Antigen–antibody complexes were then visualized by ECL system (Vilber Lourmat, Collegien, France) using an enhance chemiluminescence detection kit (Amersham Bioscience, Piscataway, NJ, USA). Band intensity was analyzed using ImageJ software (National Institutes of Health, Maryland, USA). 

### 4.13. Statistical Analysis

All results are presented as mean ± standard error of the mean (SEM). One-way analysis of variance (ANOVA) followed by Tukey’s post hoc test (SPSS version 22; IBM, Armonk, NY, USA) was used to determine statistical significance. A *p*-value <0.05 was considered statistically significant. Graph analyses were performed using GraphPad PRISM 6.0 (GraphPad Software, San Diego, CA, USA). 

## 5. Conclusions

In conclusion (Figure 5), our results showed that SC could upregulate sperm and decrease apoptosis of spermatogenic cells. Varicocele-induced testicular dysfunction is associated with ROS accumulation, release of proinflammatory cytokines, ER stress, and the mitochondrial signaling pathway. SC can protect germ cell apoptosis mediated testicular dysfunction via crosstalk between oxidative stress/ER stress and the mitochondrial signaling pathway.

## Figures and Tables

**Figure 1 ijms-20-05785-f001:**
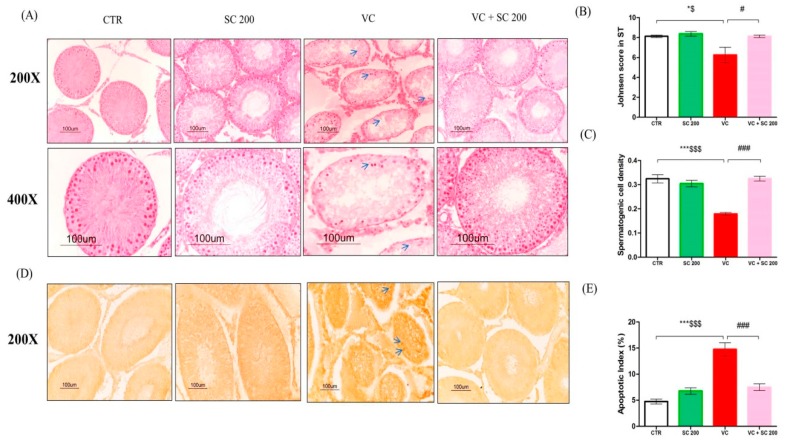
Effects of SC extract on histological changes and apoptotic index in varicocele (VC)-induced testes tissues of male Sprague–Dawley (SD) rats. (**A**) Hematoxylin and eosin staining. Blue arrows indicate vaculolation and decrease of germ cells in seminiferous tubules. (**B**) Johnsen score compared among different groups. (**C**) Spermatogenic cell density of seminiferous tubules. (**D**) TUNEL staining. Blue arrows indicate TUNEL-positive cell stained with dark brown color. (**E**) Apoptotic index. Data are presented as mean ± SEM (*n* = 10). Statistical analyses were performed using one-way ANOVA followed by Tukey’s post hoc test. * *p* < 0.05 vs. CTR group, ** *p* < 0.01 vs. CTR group, *** *p* < 0.001 vs. CTR group, ^#^
*p* < 0.05 vs. VC group, ^##^
*p* < 0.01 vs. VC group and ^###^
*p* < 0.001 vs. VC group, ^$^
*p* < 0.05 vs. SC group, ^$$^
*p* < 0.01 vs. SC group and ^$$$^
*p* < 0.001 vs. SC group. CTR, control; SC 200, SC 200 mg/kg p.o; VC, varicocele; VC + SC 200, SC 200 mg/kg; SC, *Schisandra chinensis*; p.o., per oral; ANOVA, analysis of variance; SEM: standard error of the mean.

**Figure 2 ijms-20-05785-f002:**
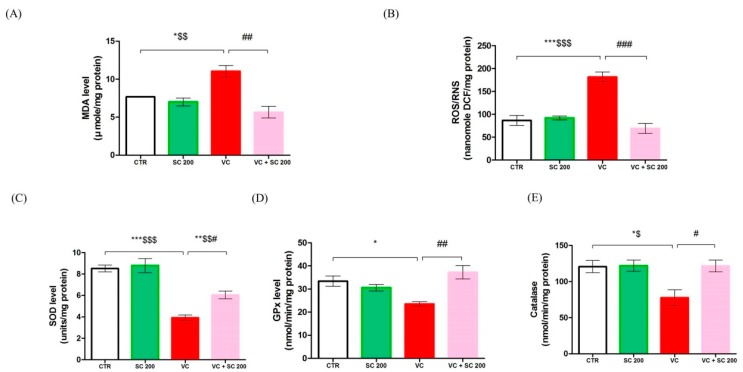
Effects of SC extract on biomarkers of oxidative stress in VC-induced testes tissues of male SD rats. (**A**) Malondehyde level, (**B**) reactive oxygen species (ROS)/reactive nitrogen species (RNS) level, (**C**) superoxide dismutase (SOD) level, (**D**) glutathione peroxidase (GPx) level, (**E**) catalase level. Data are presented as mean ± SEM (*n* = 10). Statistical analyses were performed using one-way ANOVA followed by Tukey’s post hoc test. * *p* < 0.05 vs. CTR group, ** *p* < 0.01 vs. CTR group, *** *p* < 0.001 vs. CTR group, ^#^
*p* < 0.05 vs. VC group, ^##^
*p* < 0.01 vs. VC group and ^###^
*p* < 0.001 vs. VC group, ^$^
*p* < 0.05 vs. SC group, ^$$^
*p* < 0.01 vs. SC group and ^$$$^
*p* < 0.001 vs. SC group. CTR, control; SC 200, SC 200 mg/kg p.o; VC, varicocele; VC + SC 200, SC 200 mg/kg; SC, *Schisandra chinensis*; p.o., per oral; ANOVA, analysis of variance; SEM: standard error of the mean.

**Figure 3 ijms-20-05785-f003:**
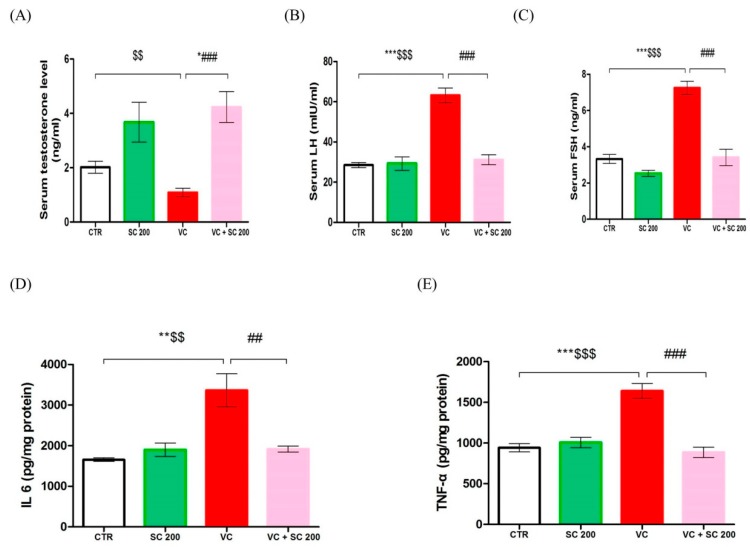
Effects of SC extract on serum hormone level and testes tissues inflammatory biomarkers in VC-induced male SD rats. (**A**) Serum testosterone level, (**B**) serum luteinizing hormone (LH) level, (**C**) serum follicle stimulating hormone (FSH) level, (**D**) IL-6 level in testicular tissue, (**E**) TNF-α level in testicular tissue. Data are presented as mean ± SEM (*n* = 10). Statistical analyses were performed using one-way ANOVA followed by Tukey’s post hoc test. * *p* < 0.05 vs. CTR group, ** *p* < 0.01 vs. CTR group, *** *p* < 0.001 vs. CTR group, ^#^
*p* < 0.05 vs. VC group, ^##^
*p* < 0.01 vs. VC group and ^###^
*p* < 0.001 vs. VC group, ^$^
*p* < 0.05 vs. SC group, ^$$^
*p* < 0.01 vs. SC group and ^$$$^
*p* < 0.001 vs. SC group. CTR, control; SC 200, SC 200 mg/kg p.o; VC, varicocele; VC + SC 200, SC 200 mg/kg; SC, *Schisandra chinensis*; p.o., per oral; ANOVA, analysis of variance; SEM: standard error of the mean.

**Figure 4 ijms-20-05785-f004:**
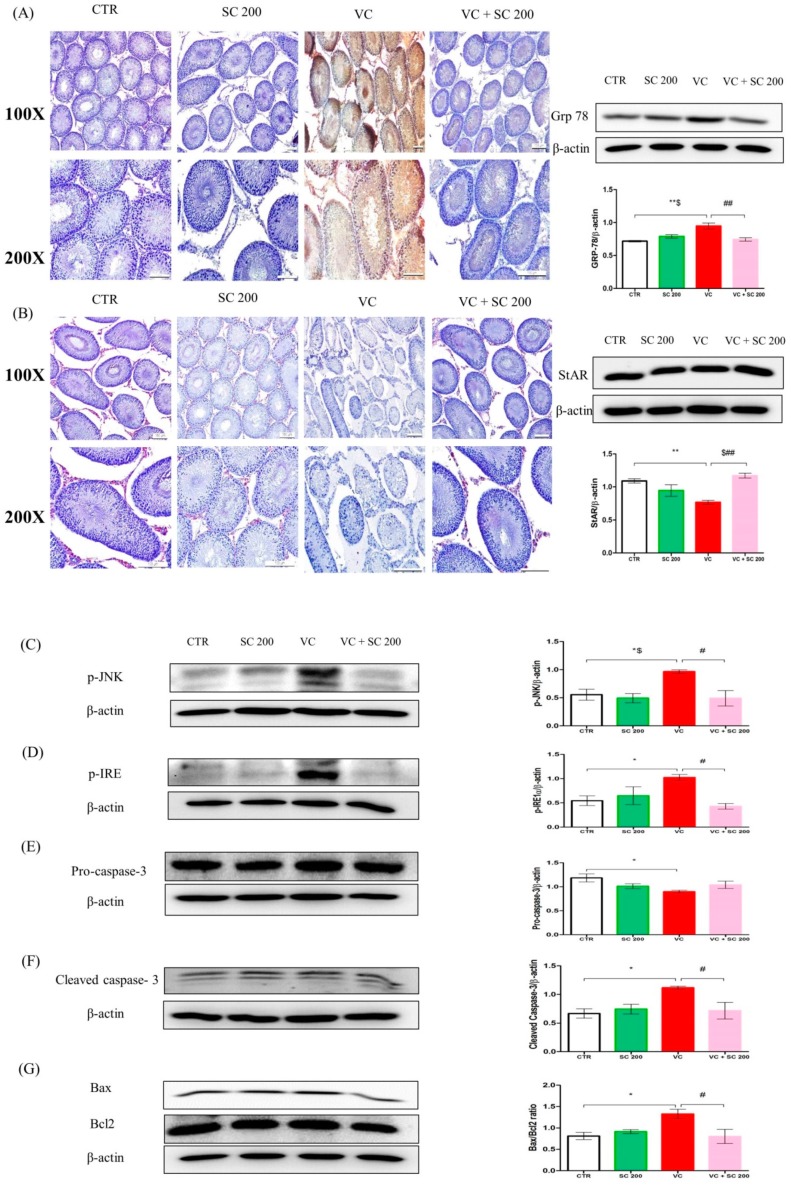
SC extract upregulates protein expression levels of markers of ER stress, apoptosis, and StAR protein in VC-induced testis tissues of male SD rats. (**A**) Grp 78 determined by Western blot and immunohistochemistry staining. In immunohistochemical staining, weak stains were found in the control group and VC + SC 200 group. Strong dark brown staining was found in the VC group. High expression levels of Grp 78 were detected in the VC group by Western blot. (**B**) StAR protein level determined by Western blot and immunohistochemistry staining. In immunohistochemical staining, strong red color stains were found in Leydig cells of the control group and the VC + SC 200 group, while weak staining was noted in the VC group. Decrease in expression of StAR protein in VC was noted by Western blot. (**C**) Phosphorylated c-Jun-N-terminal kinase (p-JNK), (**D**) p-IRE, (**E**) pro-caspase 3, (**F**) Cleaved caspase 3, (**G**) Bax:Bcl2 ratio. Data are presented as mean ± SEM (*n* = 10). Statistical analyses were performed using one-way ANOVA followed by Tukey’s post hoc test. * *p* < 0.05 vs. CTR group, ** *p* < 0.01 vs. CTR group, *** *p* < 0.001 vs. CTR group, ^#^
*p* < 0.05 vs. VC group, ^##^
*p* < 0.01 vs. VC group and ^###^
*p* < 0.001 vs. VC group, ^$^
*p* < 0.05 vs. SC group, ^$$^
*p* < 0.01 vs. SC group and ^$$$^
*p* < 0.001 vs. SC group. CTR, control; SC 200, SC 200 mg/kg p.o; VC, varicocele; VC + SC 200, SC 200 mg/kg; SC, *Schisandra chinensis*; p.o., per oral; ANOVA, analysis of variance; SEM: standard error of the mean.

**Figure 5 ijms-20-05785-f005:**
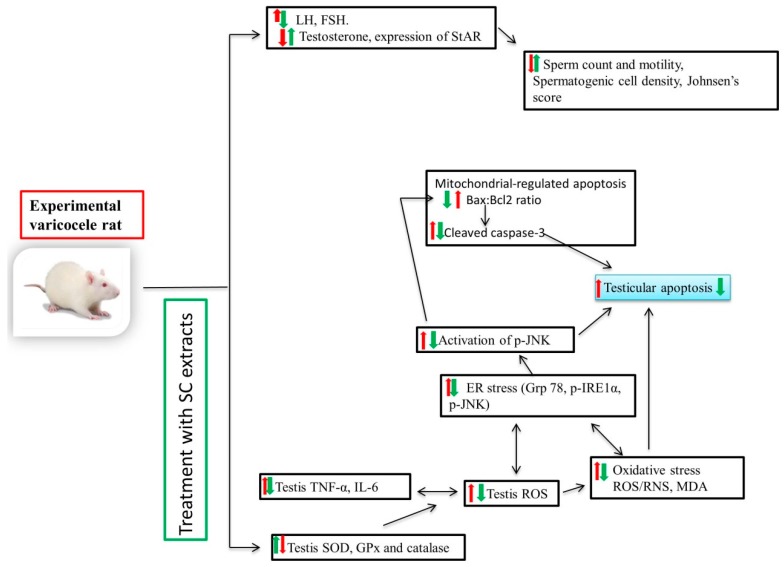
Schematic diagram showing molecular mechanism of SC in varicocele-induced testicular dysfunction in male SD rats. ER: endoplasmic reticulum; ROS/RNS: reactive oxygen species/reactive nitrogen species; MDA: malondialdehyde; SOD: superoxide dismutase; GPx: glutathione peroxidase; IL-6: interleukin-6; TNF-α: tumor necrosis factor-α; GRP-78: glucose-regulated protein-78; p-JNK: phosphorylated c-Jun-N-terminal kinase; p-IRE1α): phosphorylated inositol-requiring transmembrane kinase/endoribonuclease 1α; JNK: C-jun-N-terminal kinase; Bax: BCL 2-associated X protein; Bcl-2: B-cell lymphoma 2; StAR: steroidogenic acute regulatory protein.

**Figure 6 ijms-20-05785-f006:**
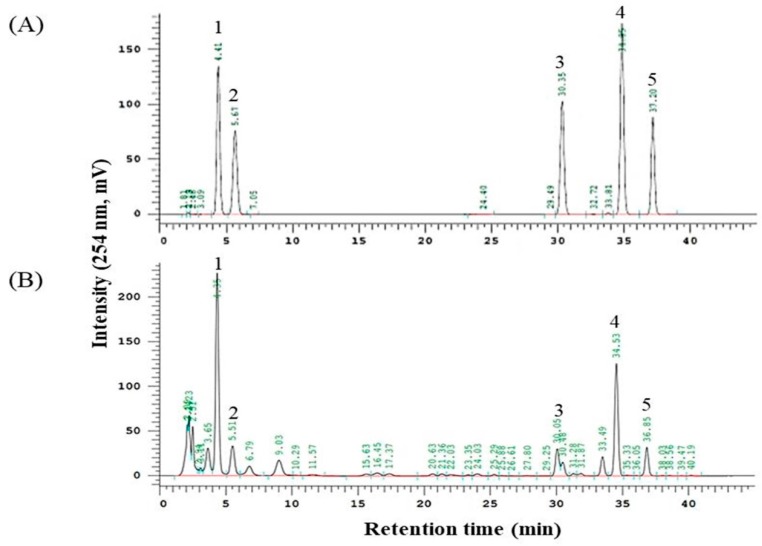
HPLC chromatogram of *Schisandra chinensis* Baillon. (**A**) lignan standards and (**B**) *Schisandra chinensis* Baillon Peaks: Schisandrol A (1), Schisandrol B (2), Schisandrin A (3), Gomisin N (4), and Schisandrin C (5).

**Table 1 ijms-20-05785-t001:** The effect of *Schisandra chinensis* Baillon (SC) extract on body weight and reproductive organ weight in VC-induced male SD rats.

Parameters	CTR	SC 200	VC	VC + SC 200
Body weight (sacrifice; g)	413.50 ± 4.99	426.50 ± 6.69	416.00 ± 4.02	423.40 ± 7.64
Testis weight (g)	2.06 ± 0.03	2.14 ± 0.04	1.77 ± 0.12 *^$$^	2.06 ± 0.03 ^#^
Epididymis weight (g)	0.72 ± 0.03.	0.76 ± 0.01	0.68 ± 0.02	0.74 ± 0.01
Prostate weight (g)	1.01 ± 0.03	1.09 ± 0.04	1.10 ± 0.08	1.13 ± 0.02
Seminal Vesicle weight (g)	1.36 ± 0.04	1.23 ± 0.04	1.27 ± 0.05	1.24 ± 0.03
Penis weight (g)	0.35 ± 0.01	0.32 ± 0.01	0.33 ± 0.01	0.33 ± 0.01
Kidney weight (g)	1.31 ± 0.01	1.37 ± 0.02	1.31 ± 0.02	1.35 ± 0.04

Data were presented as mean ± SEM, *n* = 10. Statistical analyses were performed using one-way ANOVA followed by Tukey’s post hoc test. * *p* < 0.05 vs. CTR group, ^#^
*p* < 0.05 vs. VC group, ^$$^
*p* < 0.01 vs. SC group. CTR, control; SC 200, SC 200 mg/kg p.o; VC, varicocele; VC + SC 200, SC 200 mg/kg; SC, *Schisandra chinensis*; p.o., per oral; ANOVA, analysis of variance; SEM: standard error of the mean.

**Table 2 ijms-20-05785-t002:** The effect of SC extract on sperm count and motility of vas deferens and epididymis.

Parameters	CTR	SC 200	VC	VC + SC 200
Sperm count (10^6^/mL)				
Vas deferens	30.65 ± 1.67	30.00 ± 2.23	18.25 ± 2.72 **^$$^	33.75 ± 1.56 ^###^
Epididymis	45.15 ± 1.31	44.05 ± 1.71	32.18 ± 3.23 **^$$^	43.80 ± 1.50 ^##^
Sperm motility (%)				
Vas deferens	59.42 ± 3.06	57.78 ± 2.74	34.65 ± 3.23 ***^$$$^	58.85 ± 4.08 ^###^
Epididymis	33.57 ± 5.03	42.23 ± 1.90	22.21 ± 2.03 *^$$$^	42.65 ± 1.35 ^###^

Data are presented in mean ± SEM, *n* = 10. Statistical analyses were performed using one-way ANOVA followed by Tukey’s post hoc test. * *p* < 0.05 vs. CTR group, ** *p* < 0.01 vs. CTR group, *** *p* < 0.001 vs. CTR group, ^##^
*p* < 0.01 vs. VC group and ^###^
*p* < 0.001 vs. VC group, ^$$^
*p* < 0.01 vs. SC group and ^$$$^
*p* < 0.001 vs. SC group. CTR, control; SC 200, SC 200 mg/kg p.o; VC, varicocele; VC + SC 200, SC 200 mg/kg; SC, *Schisandra chinensis*; p.o., per oral; ANOVA, analysis of variance; SEM: standard error of the mean.

**Table 3 ijms-20-05785-t003:** Effect of SC on sperm motility.

Group		Zero		3 h		Increase in Sperm Motility (%)
		Count (10^6^/mL)	Motility (%)	Count (10^6^/mL)	Motility (%)	
Patient 1	Control	38	57.8	32	50.0	–13.5
	SC 0.05 mg/mL	38	50.0	32	66.5	33.0
Patient 2	Control	62	67.0	58	65.7	–1.9
	SC 0.05 mg/mL	66	64.2	54	74.5	16.0
Patient 3	Control	31	51.6	29	38.0	–26.4
	SC 0.05 mg/mL	34	45.8	23	48.5	5.9

SC, Schisandra chinensis.

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
