# Peer review of "The Effect of Schisandra chinensis Baillon on Cross-Talk between Oxidative Stress, Endoplasmic Reticulum Stress, and Mitochondrial Signaling Pathway in Testes of Varicocele-Induced SD Rat"

_ijms, 2019, doi:10.3390/ijms20225785_

Round 1

Reviewer 1 Report

The study “The effect of Schisandra chinensis Baillon on cross-talk between oxidative stress, endoplasmic reticulum stress, and mitochondrial signaling pathway in testes of varicocele-induced SD rat” by Karna et al have shown that, Schisandra chinensis Baillon (SC) fruit extract (traditional Chinese medicinal plant) reversed the oxidative stress, ER stress and Germ cell apoptosis in testes of varicocele-induced SD rat. Although, authors have presented an interesting finding in their manuscript, however, their experimental data qualitatively are not suitable for publication in this present form. Therefore, this manuscript needs additional experimental support to draw the conclusion. 1. Several reports reveal that SC contains many bioactive compounds, including lignans, triterpenes, phenolic acids, flavonoids and polysaccharides. Percentage of these bioactive compounds varies during crude extract preparation. Therefore, authors should include the analysis data of major active compound present in their crude extract prepared from SC fruit in order to draw their conclusion. 2. It is not clear how did the authors select the dose of SC (200 mg/kg in vivo dose and 0.05 mg/ml ex vivo dose) in their experiment. 3. The result section is not written properly. Authors should include the rationale behind the each experimental design. 3. Minor check: Authors needs to check Page 2, line 74 (Major ligands of S. chinensis fruits….) “ligands” may be replaced with “lignans”.

Author Response

Dear reviewer,

We are pleased to submit our revised manuscript. WE are thankful for your detail review and kind comments. We have addressed all the comments raised by you. 

We look forward to your response.

Thanking you,

Dr. Jung Kwan Park

Reviewer 2 Report

The manuscript submitted by Karna et al., entitled “The effect of Schisandra chinensis Baillon on cross-2 talk between oxidative stress, endoplasmic reticulum 3 stress, and mitochondrial signaling pathway in testes 4 of varicocele-induced SD rat” demonstrated that Schisandra chinensis Baillon (SC) protected the varicocele (VC)-induced testicular dysfunction and germ cell apoptosis involving ER-stress and mitochondrial apoptotic pathway. The manuscript written well and the experimental set up and experiments were designed well. However, the authors need to address the following concern about this manuscript.

Major comments

What is the active component of Schisandra chinensis Baillon (SC) protecting the testis dysfunction and sperm motility? Did the specific component of SC has been reported earlier? In Fig. 4, the western blot results of procaspase, BCL2, Bax were not resembles the author’s conclusion and the bar graph data. Need to change and give good quality convincing blots. Did the authors checked PERK level (one the UPR responsive gene that is involved in BIP activation and CHOP pathway)? It would be interesting to look into that also. The authors only checked Caspase 3, BCl2/BAX and involving mitochondria, need more experiments to conclude this phenomenon. Like cytochrome release, mitochondrial pore formation assay Did mitochondrial ROS/ antioxidant pathway involved in Varicocele mediated testicular dysfunction and SC mediated protection? The authors needs to discuss how the SC extracts increases the sperm motility in patients samples? Direct exposure of sperm to SC increases sperm motility? Means whether it prevent testicular damage thereby prevent sperm damage or directly involves in sperm health?

Author Response

Dear reviewer,

We are pleased to submit our comments raised by you. We are thankful for your detailed review and kind comments.

Reviewer# 2: (Comments to the Author):

Comment #1

What is the active component of Schisandra chinensis Baillon (SC) protecting the testis dysfunction and sperm motility? Did the specific component of SC has been reported earlier?

Reply 1:

Schisandra chinensis Baillon has been traditionally used for the treatment of male infertility. Previous study by Zhang et al 2013 reported the protective effect of Schisandra chinensis Baillon (SC) polysaccharide against testis dysfunction, and improves sperm count and motility. In another study SC with other Korean traditional medicine report to improve sperm motility. However, active component of SC to improve fertility hasn’t reported yet. From our study Schisandrol A showed to improve sperm motility in human semen sample.

Zhang, Y.; Shen, N.; Qi, L.; Chen, W.; Dong, Z.; Zhao, D. H., [Efficacy of Schizandra chinesis polysaccharide on cyclophosphamide induced dyszoospermia of rats and its effects on reproductive hormones]. Zhongguo Zhong Xi Yi Jie He Za Zhi 2013, 33, (3), 361-4.

Zhou SH, Deng YF, Weng ZW, Weng HW, Liu ZD. Traditional Chinese Medicine as a Remedy for Male Infertility: A Review. World J Mens Health 2019, 37(2):175-85.

Bae WJ, Ha US, Kim KS, Kim SJ, Cho HJ, Hong SH, Lee JY, Wang Z, Hwang SY, Kim SW. Effects of KH-204 on the expression of heat shock protein 70 and germ cell apoptosis in infertility rat models. BMC Complement Altern Med 2014, 14:367.

Jo J, Lee SH, Lee JM, Jerng UM. Semen Quality Improvement in a Man with Idiopathic Infertility Treated with Traditional Korean Medicine: A Case Report. Explore (NY) 2015, 11(4):320-3.

Comment #2

In Fig. 4, the western blot results of procaspase, BCL2, Bax were not resembles the author’s conclusion and the bar graph data. Need to change and give good quality convincing blots.

Reply 2:

Thank you for your detailed review and kind comments. As per reviewer comments western blot results of procaspase 3, Bcl2 and Bax blot results has been changed.

Comment #3

Did the authors checked PERK level (one the UPR responsive gene that is involved in BIP activation and CHOP pathway)? It would be interesting to look into that also.

Reply 2:

ER stress triggers apoptosis via three signaling pathways: IRE1, ATF 6 and PERK. Prolonged ER stress was associated with apoptosis and mediated primarily via PERK and IRE1 signaling pathway. In the present study, we investigated IRE1 pathway based on a previous report of IRE1-p-JNK pathway-mediated testicular apoptosis in VC-induced rat. Activation of JNK promotes translocation of Bax from cytosol to mitochondria and plays an important role in the release of cytochrome c from mitochondrial inner membrane into the cytosol and subsequent apoptosis. Furthermore, ER stress chaperone GRP 78 predominantly occurs in pachytene spermatocytes, suggesting that ER stress signaling plays an indispensable role in spermatogenesis.

Walter P, Ron D. The unfolded protein response: from stress pathway to homeostatic regulation. Science 2011, 334(6059):1081-6.

Soni KK, Zhang LT, Choi BR, Karna KK, You JH, Shin YS, Lee SW, Kim CY, Zhao C, Chae HJ et al. Protective effect of MOTILIPERM in varicocele-induced oxidative injury in rat testis by activating phosphorylated inositol requiring kinase 1alpha (p-IRE1alpha) and phosphorylated c-Jun N-terminal kinase (p-JNK) pathways. Pharm Biol 2018, 56(1):94-103.

Huo R, Zhu YF, Ma X, Lin M, Zhou ZM, Sha JH. Differential expression of glucose-regulated protein 78 during spermatogenesis. Cell Tissue Res 2004, 316(3):359-67.

Dhanasekaran DN, Reddy EP. JNK signaling in apoptosis. Oncogene 2008, 27(48):6245-51.

Comment #4

The authors only checked Caspase 3, BCl2/BAX and involving mitochondria, need more experiments to conclude this phenomenon. Like cytochrome release, mitochondrial pore formation assay.

Reply 4:

Thank you for your detailed review and kind comments. Current study proposed apoptosis related varicocele infertility and proposed different associated mechanism such as inflammation, oxidative stress, ER stress and intrinsic apoptosis pathway, and efficacy of SC in Varicocele rat. Acceleration of Bcl2 in germ cell favors cell survival, whereas elevation of Bax accelerates cell death. Present study showed similar finding and consist with previously reported in varicocele rat. Thank you for your suggestion for cytochrome release and mitochondrial pore formation assay. However, the current data suggest the ER stress pathway and its correlation with inflammation, oxidative stress and intrinsic pathway. Cytochrome release and mitochondrial pore formation assay is another marker for mitochondrial apoptosis. Current study reported Bax, Bcl2 ratio, cleaved caspase-3, procaspase-3 as the markers of intrinsic apoptosis pathway. As per your suggestion we will study the effect of SC and its relation with mitochondrial pore formation and cytochrome release mechanism in detail in our next study due to lack of project fund for this study.

Lee JD, Lee TH, Cheng WH, Jeng SY. Involved intrinsic apoptotic pathway of testicular tissues in varicocele-induced rats. World J Urol 2009, 27(4):527-32.

Onur R, Semercioz A, Orhan I, Yekeler H. The effects of melatonin and the antioxidant defence system on apoptosis regulator proteins (Bax and Bcl-2) in experimentally induced varicocele. Urol Res 2004, 32(3):204-8.

Comment #5

Did mitochondrial ROS/ antioxidant pathway involved in Varicocele mediated testicular dysfunction and SC mediated protection?

Reply 5:

Thank you for your detailed review and kind comments. Previous studies have reported the varicocele-induced testicular dysfunction associated male infertility and the role of ROS/antioxidant pathway. Several empirical therapies with herbal agents for treating VC-induced infertility have been reported, none of them has been proven to be superior to others. We are the first to report protective role of SC on varicocele-induced male infertility.

Mendes, T. B.; Paccola, C. C.; de Oliveira Neves, F. M.; Simas, J. N.; da Costa Vaz, A.; Cabral, R. E.; Vendramini, V.; Miraglia, S. M., Resveratrol improves reproductive parameters of adult rats varicocelized in peripuberty. Reproduction 2016, 152, (1), 23-35.

Moshtaghion, S. M.; Malekinejad, H.; Razi, M.; Shafie-Irannejad, V., Silymarin protects from varicocele-induced damages in testis and improves sperm quality: evidence for E2f1 involvement. Syst Biol Reprod Med 2013, 59, (5), 270-80.

Tian, R. H.; Ma, M.; Zhu, Y.; Yang, S.; Wang, Z. Q.; Zhang, Z. S.; Wan, C. F.; Li, P.; Liu, Y. F.; Wang, J. L.; Liu, Y.; Yang, H.; Zhang, Z. Z.; Liu, L. H.; Gong, Y. H.; Li, F. H.; Hu, H. L.; He, Z. P.; Huang, Y. R.; Li, Z., Effects of aescin on testicular repairment in rats with experimentally induced varicocele. Andrologia 2014, 46, (5), 504-12.

Asadi, N.; Kheradmand, A.; Gholami, M.; Saidi, S. H.; Mirhadi, S. A., Effect of royal jelly on testicular antioxidant enzymes activity, MDA level and spermatogenesis in rat experimental Varicocele model. Tissue Cell 2019, 57, 70-77.

 Comment #6

The authors need to discuss how the SC extracts increases the sperm motility in patient’s samples? Direct exposure of sperm to SC increases sperm motility? Means whether it prevents testicular damage thereby prevent sperm damage or directly involves in sperm health?

Reply 6:

Thank you for your detailed review and kind comments. As per reviewer comment we have added supplementary Figure 5 to support increase in patient sample. Sperm motility showed increase in major compound of SC, Schisandrol A (schisandrin). Schisandrol A has antioxidant activities. SC lignans may protect spermatozoa by keeping enzymatic and antioxidant process such as ROS in optimum condition. SC extract might exhibit protective effects on sperm motility via regulation of the proton channel, which we have planed of study in our next project due to lack of fund. Result of our study in patient sample supports the antioxidant activities of SC extract in VC-induced SD rats.

Thanking you,

Dr. Jong Kwan Park

Round 2

Reviewer 1 Report

Authors have addressed all the comments. Therefore, the manuscript can be accepted in this present from.